# A Simplified Genomic Profiling Approach Predicts Outcome in Metastatic Colorectal Cancer

**DOI:** 10.3390/cancers11020147

**Published:** 2019-01-27

**Authors:** Carlo Capalbo, Francesca Belardinilli, Domenico Raimondo, Edoardo Milanetti, Umberto Malapelle, Pasquale Pisapia, Valentina Magri, Alessandra Prete, Silvia Pecorari, Mariarosaria Colella, Anna Coppa, Caterina Bonfiglio, Arianna Nicolussi, Virginia Valentini, Alessandra Tessitore, Beatrice Cardinali, Marialaura Petroni, Paola Infante, Matteo Santoni, Marco Filetti, Valeria Colicchia, Paola Paci, Silvia Mezi, Flavia Longo, Enrico Cortesi, Paolo Marchetti, Giancarlo Troncone, Diana Bellavia, Gianluca Canettieri, Giuseppe Giannini

**Affiliations:** 1Department of Molecular Medicine, University La Sapienza, 00161 Rome, Italy; carlo.capalbo@uniroma1.it (C.C.); francesca.belardinilli@uniroma1.it (F.B.); domenico.raimondo@uniroma1.it (D.R.); colella.1624422@studenti.uniroma1.it (M.C.); virginia.valentini@uniroma1.it (V.V.); valeria.colicchia@uniroma1.it (V.C.); diana.bellavia@uniroma1.it (D.B.); 2Department of Medical Oncology Sant’ Andrea Hospital, I-00189 Rome, Italy; marco.filetti@uniroma1.it (M.F.); paolo.marchetti@uniroma1.it (P.M.); 3Department of Physics, University La Sapienza, 00185 Rome, Italy; edoardo.milanetti@uniroma1.it; 4Department of Public Health, University Federico II, 80131 Naples, Italy; umberto.malapelle@unina.it (U.M.); pasquale.pisapia@unina.it (P.P.); giancarlo.troncone@unina.it (G.T.); 5Department of Radiological Oncological and Pathological Sciences, University La Sapienza, 00161 Rome, Italy; valentina.magri@uniroma1.it (V.M.); alessandra.prete@uniroma1.it (A.P.); silvia.pecorari@uniroma1.it (S.P.); silvia.mezi@uniroma1.it (S.M.); flavia.longo@uniroma1.it (F.L.); enrico.cortesi@uniroma1.it (E.C.); 6Department of Experimental Medicine, University La Sapienza, 00161 Rome, Italy; anna.coppa@uniroma1.it (A.C.); arianna.nicolussi@uniroma1.it (A.N.); 7National Institute of Gastroenterology-Research Hospital, IRCCS “S. de Bellis”, Castellana Grotte, 70013 Bari, Italy; catia.bonfiglio@irccsdebellis.it; 8Department of Biotechnological and Applied Clinical Sciences, University of L’Aquila, 67100 L’Aquila, Italy; alessandra.tessitore@univaq.it; 9Institute of Cell Biology and Neurobiology, National Research Council, Campus A. Buzzati-Traverso, 00015 Monterotondo Scalo, Italy; beatrice.cardinali@cnr.it; 10Center for Life Nano Science@Sapienza, Istituto Italiano di Tecnologia, 00161 Rome, Italy; marialaura.petroni@iit.it (M.P.); paola.infante@iit.it (P.I.);; 11Oncology Unit, Macerata Hospital, 62012 Macerata, Italy; mattymo@alice.it; 12Institute for Systems Analysis and Computer Science “Antonio Ruberti”, National Research Council, 00185 Rome, Italy; paola.paci@iasi.cnr.it; 13Pasteur Institute-Cenci Bolognetti Foundation, 00161 Rome, Italy

**Keywords:** precision medicine, predictive, NGS, genomic profiling, chemotherapy

## Abstract

The response of metastatic colorectal cancer (mCRC) to the first-line conventional combination therapy is highly variable, reflecting the elevated heterogeneity of the disease. The genetic alterations underlying this heterogeneity have been thoroughly characterized through omic approaches requiring elevated efforts and costs. In order to translate the knowledge of CRC molecular heterogeneity into a practical clinical approach, we utilized a simplified Next Generation Sequencing (NGS) based platform to screen a cohort of 77 patients treated with first-line conventional therapy. Samples were sequenced using a panel of hotspots and targeted regions of 22 genes commonly involved in CRC. This revealed 51 patients carrying actionable gene mutations, 22 of which carried druggable alterations. These mutations were frequently associated with additional genetic alterations. To take into account this molecular complexity and assisted by an unbiased bioinformatic analysis, we defined three subgroups of patients carrying distinct molecular patterns. We demonstrated these three molecular subgroups are associated with a different response to first-line conventional combination therapies. The best outcome was achieved in patients exclusively carrying mutations on *TP53* and/or *RAS* genes. By contrast, in patients carrying mutations in any of the other genes, alone or associated with mutations of *TP53/RAS*, the expected response is much worse compared to patients with exclusive *TP53/RAS* mutations. Additionally, our data indicate that the standard approach has limited efficacy in patients without any mutations in the genes included in the panel. In conclusion, we identified a reliable and easy-to-use approach for a simplified molecular-based stratification of mCRC patients that predicts the efficacy of the first-line conventional combination therapy.

## 1. Introduction

Metastatic Colorectal Cancer (mCRC) is a major burden worldwide, with a 5-year survival rate of about 13% [1,2]. Currently, standard chemotherapy includes oxaliplatin-based (FOLFOX) and irinotecan-based (FOLFIRI) regimens, which appear equivalent in efficacy and activity [3,4] allowing a median overall survival (OS) of about 18–20 months. A significant therapeutic improvement in the upfront treatment of mCRC can be achieved by a “triplet” approach consisting of 5-FU, platinum and irinotecan-based chemotherapy (FOLFOXIRI) [5]. The addition of targeted drugs, such as monoclonal antibodies directed against VEGF (Bevacizumab) or EGFR (Cetuximab and Panitumumab) have increased the median OS up to 30 months [6,7,8,9]. Importantly, *KRAS* and *NRAS* (*RAS*) gene mutations confer innate resistance to anti-EGFR therapy [10,11]. Consequently, screening for *RAS* mutations is now mandatory for first-line therapeutic decisions. Of interest, two large studies (FIRE-3 and PEAK) indicated the absence of significant differences in the outcome of *RAS* wild-type mCRC patients undergoing first-line treatment with either bevacizumab or anti-EGFR added to FOLFOX or FOLFIRI regimens [12,13]. Thus, while the choice of the most appropriate first line therapy is extremely important for the general outcome of any cancer patient, available treatments appear substantially equivalent for *RAS* wild-type mCRC patients in the clinical routine, where decisions also need to take into account the patient’s clinical features (performance, age and comorbidities), toxicity issues and treatment intent. To this end, an evidence-based algorithm for first-line chemotherapy decision-making in mCRC has been proposed [14].

Of relevance, the response to these standard therapeutic approaches differs from patient to patient, largely reflecting the clinical heterogeneity of the disease, and leading to a yet unsatisfactory OS rate of mCRC patients. This is most likely related to the broad intertumor and intratumor molecular heterogeneity, which has been ascertained by global and integrated Next Generation Sequencing (NGS)-based genomic and transcriptomic profiling [15,16,17,18]. To rationalize this heterogeneity, these efforts have recently converged on the definition four main consensus molecular subgroups (CMSs): CMS1 (MSI immune); CMS2 (canonical); CMS3 (metabolic) and CMS4 (mesenchymal) [16], each of which contains multiple actionable targets.

The recent experience with anti-EGFR treatment and the poor response to PI3K inhibitors [19] clearly suggests that a better stratification of patients according to specific molecular biomarkers may dramatically improve the efficacy of targeted therapies, and perhaps also chemotherapy. Nonetheless, beside *RAS* gene mutations, and *BRAF*^V600E^ mutation (for which a prognostic and a predictive significance have been established [20,21,22]) the prognostic/predictive value of mutations in other genes (i.e., *TP53*, or *PIK3CA*) is still unclear and no predictive markers are yet available for the response to VEGF inhibitors [23,24].

Thus, despite the substantial advances in understanding the molecular basis of this disease, how to use the molecular subtyping to guide clinical approaches still appears unclear at the moment and how to use the information on the molecular complexity of each CRC patients for their clinical management has been addressed poorly in current literature.

We and others have recently shown that multigene panel sequencing, including, but not limited to, clinically relevant *RAS* and *BRAF* hot spots, is a valid, flexible, sensitive and economical method for the routine diagnostics of mCRC, which may also provide additional information with no extra costs [25,26,27,28,29]. As a first result of its implementation in the clinical routine for mCRC patients, we report here on the identification of patterns of molecular alterations predictive of the response to standard first-line therapies.

## 2. Results

We investigated whether clinical sequencing with a multigene panel of 22 genes might contribute to a better molecular characterization and/or improved stratification of mCRC patients.

Our cohort of 77 mCRC patients with features reported in Table 1 had been treated with first line conventional combination therapy, consisting of FOLFOX/CAPEOX, FOLFIRI or FOLFOXIRI, eventually associated with monoclonal antibody-mediated targeting of either EGFR or VEGF, depending on *RAS* mutational status (Table 1). Patients unfit for combination therapies, because of their comorbidities and/or age (PS ≥ 2), were treated with a therapy adapted to their clinical condition.

We screened this cohort for the presence/absence of mutations in a panel of twenty-two genes, designed with the OncoNetwork Consortium, including *RAS* (*KRAS*/*NRAS*), *EGFR*, *BRAF*, *PIK3CA*, *AKT1*, *ERBB2*, *PTEN*, *STK11*, *MAP2K1*, *ALK*, *DDR2*, *CTNNB1*, *MET*, *TP53*, *SMAD4*, *FBXW7*, *FGFR3*, *NOTCH1*, *ERBB4*, *FGFR1*, and *FGFR2*. The performance of this approach in our hands has been described elsewhere [25]. The mutation frequency found for each gene (Figure 1) was in line with previously reported findings [15], indicating that our relatively small cohort was well representative of the disease. Within this series, two or more mutated genes were found in 59.7% of the patients (46/77), while 27.3% of the patients (21/77) showed only one mutation. *TP53* and *RAS* mutations were the most common alterations found (57.1% and 50.6%, respectively). The mutation frequency in other genes was much lower, being *PIK3CA* the most frequently mutated (14.3%), followed by *FBXW7* (9.1%), *BRAF* (9.1%), *SMAD4* (7.8%), *MET* (6.5%), *FGFR1* (2.6%). *NOTCH1*, *STK11*, *EGFR*, *DDR2*, *CTNNB1*, *FGFR3*, *PTEN*, *ERBB4* were mutated in only 1.3% of patients.

Overall, we identified 51 patients carrying actionable gene mutations, as defined by Chakravarty et al. [30], 22 of which carried druggable alterations (Table 2). The greatest number of patients (39, 75%) carried *RAS* mutations, which guided their exclusion from anti-EGFR therapy. Five patients carried the MSI-H phenotype, predictive of response to anti-immune checkpoint therapy [31]. Importantly, the vast majority of patients positive or negative for actionable mutations also carried additional oncogenic genetic alterations, which in principle could contribute to an individual clinical variability affecting responsiveness to standard and target-driven therapies (Table 2). Overlooking this molecular complexity might be largely responsible for treatment failure, when standard or innovative targeted approaches are used.

As a first attempt to take into account this molecular complexity, we subjected our sample to an unsupervised hierarchical clustering analysis of the mutation patterns, which identified three main groups (Figure 2). With few exceptions, the first group was characterized by mutations uniquely occurring in the *TP53* and/or *RAS* genes. This group was named p53/RAS Group (PRG). The second group was characterized by complete absence of mutations in the 22 genes of the panel and was named No Mutations Group (NMG). Patients with mutations in at least one of the other genes, with or without coexisting mutations in *TP53* and/or *RAS* genes, clustered in the third group, which was named All Genes Group (AGG).

According to these molecular features, 35 patients (45%) fell into the PRG group, 32 patients (42%) into the AGG and 10 patients (13%) into the NMG (Figure 3). We then investigated whether these molecular signatures could predict the response to standard first-line therapies.

Median Progression Free Survival (PFS) in the overall population was 10 months. Of note, patients belonging to the PRG cluster had a significantly longer PFS than AGG and NMG groups, (12 months, eight months and six months, respectively; *p* = 0.0039). The log-rank test showed significant differences between PRG and AGG curves (*p* = 0.00735) and PRG vs NMG (*p* = 0.00431). In contrast AGG and NMG showed no significant differences (Figure 3). Consistently, stratification into PRG and AGG was a significant predictor of PFS both at univariate (*p* = 0.011) and multivariate (*p* = 0.039) analyses, independently of the number and nature of other variables included in the analysis, suggesting it is an independent predictive factor. NMG stratification was a significant predictor of PFS at univariate (*p* = 0.010), but it wasn’t at multivariate analyses, likely due to the low number of samples. With the exception of gender, no other factor significantly affected PFS (Table 3).

To test whether our group stratification could be useful specifically in the RAS WT subset, we further assessed its impact on PFS on the patients sharing this molecular feature. Consistent with overall data analysis, stratification into PRG, AGG and NMG was a significant predictor of PFS also in this subset (Appendix A).

## 3. Discussion

NGS is emerging as an efficient and cost-effective technology to provide information on potential clinical utility, since it can track molecular heterogeneity in cancer samples [15,16]. Several studies suggest that high-throughput NGS testing in metastatic CRC may become a comprehensive and fast approach to support personalized therapy [32,33,34] and indicate that recurrent alterations of a considerable number of pathways and genes are found in CRC. These include the WNT (APC, β-catenin, MYC), RTKs-MAPK (ERB-Bs, RAS, BRAF, MAPK), PI3K-PTEN, p53, EMT/TGF-β, mismatch repair, epithelial-mesenchymal transition, immune and inflammation pathways [15,16,35]. Nevertheless, a very limited number of molecular alterations gained relevance in the routine clinical settings for mCRC, beside *RAS*/*BRAF* mutations, and more recently MMR deficiency that predicts a positive response to the immune checkpoint inhibitor pembrolizumab [31]. As a result, only modest therapeutic advances have been made, and other molecular markers of established predictive value, to support the therapeutic choices, are currently missing.

Clinical sequencing of a limited number of genes can now be easily achieved with very reasonable turnaround time and costs on benchtop instruments [25,29]. Nonetheless, so far very few efforts have been made to take into account how the molecular complexity of mCRC impact on the efficacy of novel, but also currently established, therapeutic approaches. Here, we have used a simplified NGS-based platform to characterize a cohort of 77 mCRCs patients managed in our cancer clinic and treated according to standard guidelines for first-line conventional therapy. As expected, this approach allowed the identification of a relevant number of potentially actionable molecular alterations. Indeed, 39 patients carried RAS mutations, which guided their exclusion from anti-EGFR therapy. Twenty-two patients were carriers of druggable genetic alterations and five of the MSI-H phenotype, which could have been taken into account for personalized therapeutic choices. But most important, it also returned a representation of the molecular complexity of each tumor sample. Rather than taking into account the clinical impact of individual genomic alterations, or grouping patients based on a supervised pathway analysis [26,32,34,36,37], we decided to subject our data to an unsupervised hierarchical clustering analysis, which led us to identify three distinct molecular subgroups (PRG, AGG, NMG). The relatively small size of our cohort and the high number of different combination therapies in subsequent lines of intervention did not allow us to efficiently address the effect of this stratification on OS. Nonetheless, it revealed its utility in predicting PFS in patients treated with different first-line approaches expected to be equally effective on the base of current literature [14] and NCCN directions (Guideline for treatment of cancer by site; Colon Guideline Version 4.2018, 19 October 2018 ^©^National Comprehensive Cancer Network, www.NCCN.org).

In particular, the best outcome was achieved in PRG patients, exclusively carrying mutations on *TP53* and/or *RAS* genes. This group of patients is characterized by a minimal number of genetic alterations (19/35 carried only one mutation either on *RAS* or *TP53*). We speculate that this minimal mutation load on the RTKs-MAPK and/or p53 pathway might be responsible for a higher sensitivity to standard first-line therapy. Importantly, in the absence of additional genetic alterations, no other biological intervention could have been possible in this subset. The high prevalence of this group, representing about half of the total CRCs, probably explains why standard combination therapies have emerged over the years as the most effective approaches in mCRC patients. By contrast, the AGG molecular subgroup showed a much worse response compared to patients with exclusive *TP53*/*RAS* mutations. Importantly, 7/32 patients of this group carried the *BRAF*^V600E^ mutation, an established bad prognosis marker [22], six of which in association with other mutations and three also associated with the MSI-H phenotype.

Other 10 patients of this group had mutations in three or more targets, which can be responsible for the activation of multiple oncogenic pathways. Fourteen out of thirty-two patients of this group carried mutation on two targets, often involving either *RAS* or *TP53* in different combinations with other genes, most frequently *PIK3CA*, *SMAD4* and *MET*. Therefore, the AGG group is characterized by simultaneous activation of multiple oncogenic pathways, a condition that might reasonably confer resistance to standard first-line therapy. Importantly, 22 patients in this group were carriers of one or more druggable molecular alterations, which might have indicated the association with additional biological agents. For example, we have reported an extended survival in a *RAS* WT, *BRAF*^V600E^ mutation carrier mCRC patient treated with a combination of vemurafenib and panitumumab [20], suggesting that target-driven combination therapies might significantly improve patient’s outcome. The target-driven association of anti-EGFR, with either BRAF, PI3K or MET inhibitors could have improved PFS.

Of relevance, our molecular subgroup stratification predicted PFS also in the *RAS* WT subset of patients, where important decisions regarding the biological drug to be employed is currently dependent on the clinical status of individual patient and treatment intent, rather than molecular features [14]. Hopefully future studies will help us establish whether our stratification can provide additional hints for better therapy choices in this subset.

Finally, our data indicate that the standard first-line approaches have limited efficacy in NMG patients, characterized by the absence of mutations in any of the genes included in the panel. Of relevance, in silico analysis confirmed a similar percentage of NMG patients in the TCGA CRC dataset and demonstrated an overall reduced mutation frequency and absence of recurrent point mutations in particular oncogenic targets, suggesting that either chromosomal rearrangements and/or epigenetic events support the tumorigenic programme in this group. These latter observations suggest that alternative protocols, plausibly hitting alternative molecular mechanism of tumorigenesis, should be tested in this subgroup of patients, after appropriate molecular investigations have been carried out.

Recently, clinical implications of primary tumor side started to be analyzed in the metastatic setting, providing evidence that right-sided tumors have a worse prognosis and are less sensitive to anti-EGFR therapy [38]. Although the limited size of our cohort does not allow us to draw final conclusions about the impact of side on the predictive value of our results, it is important to recall that molecular alterations, rather than tumor location, predict outcome at a multivariate analysis in a large clinical sequencing study on CRC samples [34]. These further stresses our idea that higher attention should be conveyed on the characterization of the molecular profile of each mCRC case.

Hopefully, the examination of different and much larger molecular and clinical datasets will help us establishing whether our multigene panel method could provide predictive insights on PFS and OS for each distinct therapeutic approach, in similar or different clinical settings.

## 4. Patients and Methods

### 4.1. Patients and Data Collection

77 FFPE samples were obtained from mCRC patients managed in the clinical routine at Policlinico Umberto I Hospital (Rome). Patients’ characteristics are listed in Table 1.

Depending on their RAS mutational status, and based on individual features and on the general algorithm described in [14], patients had been treated with first line conventional combination therapy, consisting of FOLFOX/CAPEOX (33 patients), FOLFIRI (22 patients) or FOLFOXIRI (10 patients), associated with monoclonal antibodies-mediated targeting of either EGFR (18 patients) or VEGF (47 patients). Patients with ECOG performance status ≥2 and thus unfit for combination therapies, were treated with a therapy adapted to their clinical condition. 

All investigations were approved by the Ethics Committee of the University La Sapienza (Prot.: 88/18; RIF.CE:4903, 31-01-2018) and conducted according to the principles outlined in the declaration of Helsinki. 

### 4.2. DNA Extraction

The tumor cell percentage, estimated by inspecting Haematoxylin–Eosin (H&E) stained slides, ranged from 20% to 90%. The tumor area was macroscopically dissected to concentrate tumor tissue. Xylene was added once and ethanol was added twice to remove all paraffin from the tissue sample. The DNA was extracted using the QIAamp DNA FFPE Tissue kit (Qiagen, Hilden, Germany) according to the manufacturer’s instructions. Eluted DNA was quantified with Qubit 2.0 Fluorometer (Thermo Fisher Scientific, Van Allen Way, Carlsbad, CA, USA) using the Qubit dsDNA HS Assay Kit (Thermo Fisher Scientific, Eugene, OR, USA).

### 4.3. IT-PGM Sequencing and Variant Calling

IT-PGM sequencing was achieved as described [25,29]. Approximately 10 ng of DNA was required to construct barcoded and adaptor-ligated libraries using the Ion AmpliSeq Library Kit 2.0 (Thermo Fisher Scientific, Van Allen Way, Carlsbad, CA, USA) and Ion Xpress™ Barcode Adapter 1-16 Kit (Thermo Fisher Scientific). The samples were analyzed using the Ion AmpliSeq Colon and Lung Cancer Research Panel V2 (Thermo Fisher Scientific, Guilford, CT, USA) containing a single primer pool to amplify hotspots and targeted regions of 22 cancer genes frequently mutated in CRCs and NSCLCs. Templated spheres were prepared using 100 pM of each library using the Ion One Touch 2.0 machine (Thermo Fisher Scientific, Van Allen Way, Carlsbad, CA 92008, USA). Template-positive spheres were loaded into Ion chip 314 or Ion chip 316 and sequenced by IT-PGM machine (Thermo Fisher Scientific, Van Allen Way, Carlsbad, CA 92008, USA). Sequencing data were analyzed with the Ion Torrent Suite (Thermo Fisher Scientific, https://github.com/iontorrent/TS). Variants with a quality <30 were filtered out. PolyPhen-2 (Polymorphism Phenotyping v2; http://genetics.bwh.harvard.edu/pph2/), SIFT (Sorts Intolerant from Tolerant substitutions; http://provean.jcvi.org/protein_batch_submit.php?species=human) and PROVEAN (Protein Variation Effect Analyzer, http://provean.jcvi.org/protein_batch_submit.php?species=human) computational tools were used to predict the possible impact of each genetic change on the function of the corresponding protein.

### 4.4. Bioinformatic and Statistical Analysis

Mutational profile detected for each of the 77 mCRC samples was analyzed through heat-map representation. We considered a value equal to 0 for the wild-type gene and value equal to one corresponds to a mutated gene. In order to perform cluster analysis, we chose the hierarchical agglomerative clustering procedure, which has a bottom-up approach. Complete linkage method and a Euclidean distance metric for hierarchical clustering were used. This clustering method defines the distance between two clusters as the maximum distance between their individual components. At every stage of the clustering process, the two nearest clusters are merged into a new cluster. The process is repeated until the whole data set is agglomerated into one single cluster. Hierarchical clustering of samples and genes was performed with the “heatmap.2” R function by employing “gplot” packages. The Kaplan–Meier estimator was computed with “survfit” R function by employing “survival” package [39]. The log-rank test was used to test differences between groups. A Cox-regression model was applied to the data with a univariate and multivariate approach. Variables considered in the univariate analysis were mutational stratification, gender, age, tumor location, grading, adjuvant therapy, first line therapy, metastatic site, ECOG PS and surgery for the primary site.

### 4.5. Determination of MSI-H phenotype

Determination of MSI status was carried out by analysis of BAT25, BAT26, NR21, NR22 and NR24 mononucleotide repeats as previously described [40]. Briefly, one PCR primer of each pair was labelled with either FAM, HEX, or NED fluorescent markers. PCR amplification was performed under the following conditions: Denaturation at 94 °C for five minutes, 35 cycles of denaturation at 94 °C for 30 s, annealing at 55 °C for 30 s, and extension at 72 °C for 30 s. This was followed by an extension step at 72 °C for 7 min. PCR products were run on ABI PRISM 3130xl Genetic Analyzer (16 capillary DNA sequencer, Applied Biosystems). Gene Mapper software 5 (version 5.0, Applied Biosystems, Van Allen Way, Carlsbad, CA, USA) was used to calculate the size of each fluorescent PCR product. Five out of seventy-seven tumors showed MSI-H phenotype when deletions in all five mononucleotide markers were found.

## 5. Conclusions

The response of mCRC to the first-line conventional combination therapy is highly variable, reflecting the elevated heterogeneity of the disease. By a simplified NGS approach and assisted by bioinformatic analysis, we identified three molecular subgroups in a cohort of 77 mCRC patients characterized by three different mutation profiles. This new, handy, and cost-effective NGS-based stratification significantly predicts clinical responses to first-line conventional combination therapy in mCRC.

## Figures and Tables

**Figure 1 cancers-11-00147-f001:**
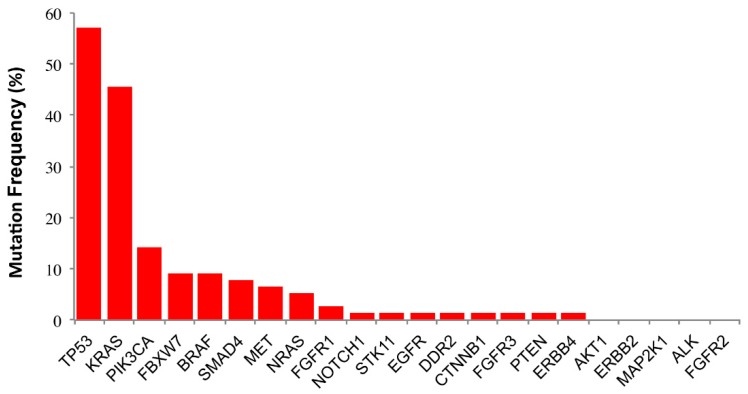
Mutation frequency for each gene of the panel in the study cohort.

**Figure 2 cancers-11-00147-f002:**
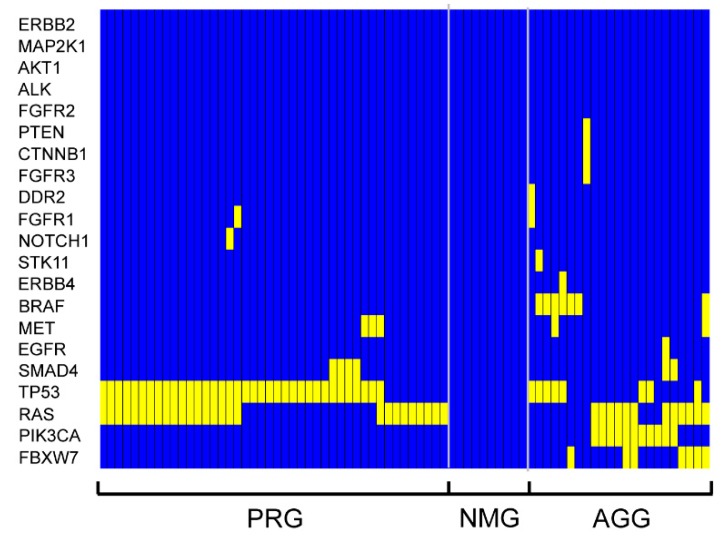
Heat-map representation of hierarchical clustering analysis of mCRC data. Columns represent the 77 samples of mCRC patients and rows represent the 22 different mutational markers. Three separate clusters were generated by this analysis. Each cluster corresponds to a distinct genetic profile. The color assigned to a cell in the heatmap grid indicates mutational condition (yellow for mutation, blue for wild-type gene) of a particular gene in a given patient sample.

**Figure 3 cancers-11-00147-f003:**
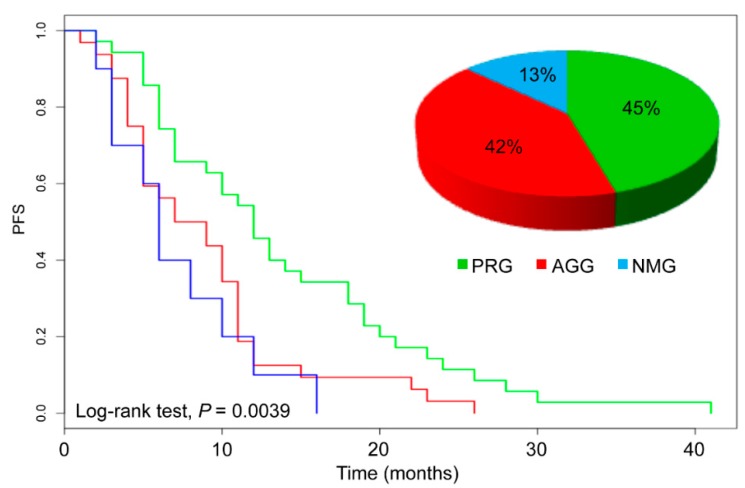
Kaplan–Meier plot showing the impact of p53/RAS Group (PRG; green), All Genes Group (AGG; red) and No Mutations Group (NMG; blue) group stratification on progression free survival (PFS) in the mCRC cohort.

**Table 1 cancers-11-00147-t001:** Characteristics of the study cohort (*n* = 77).

**Characteristics**	***n***
Sex	
Males	41
Females	36
**Age**	
Mean	64
Range	38–81
**Site**	
Right Colon	28
Left Colon	31
Rectum	18
**Metastatic site (*n*)**	
Liver	38
Peritoneum	12
Lung	17
Limph nodes	7
Bone	2
**First Line Therapy**	
Chemotherapy/anti-VEGF	47
Chemotherapy/anti-EGFR	18
Chemotherapy	12
**Performance Status ECOG**	
0–1	65
≥2	12

VEGF: Vascular Endothelial Growth Factor; EGFR: Epidermal Growth Factor Receptor; ECOG: Eastern Cooperative Oncology Group.

**Table 2 cancers-11-00147-t002:** List of the genes carrying actionable mutations in the studied metastatic colorectal cancer (mCRC) patients.

Gene ^a^	Status	No. of pts (%)	No. of pts (%) with Additional Mutation
*RAS*	WT	38 (49.4)	28 (73.7)
	Mut	39 (50.6)	31 (79.5)
*BRAF*	WT	70 (90.9)	60 (77.9)
	Mut	7 (9.1)	6 (85.7)
*EGFR*	WT	76 (98.7)	66 (85.7)
	Mut	1 (1.3)	1 (100.0)
*PIK3CA*	WT	66 (85.7)	56 (84.8)
	Mut	11 (14.3)	10 (90.9)
*MET*	WT	72 (93.5)	62 (80.5)
	Mut	5 (6.5)	5 (100.0)
*PTEN*	WT	76 (98.7)	66 (85.7)
	Mut	1 (1.3)	1 (100.0)
MSI-H ^b^	No	72 (93.5)	62 (80.5)
	Yes	5 (6.5)	5 (100.0)

^a^: Actionable genes selected according to Chakravarty et al. [30] ^b^: MSI-H status was determined with appropriate methodology as described in Materials and Methods. All other genetic alterations were detected by clinical sequencing.

**Table 3 cancers-11-00147-t003:** Univariate and multivariate analysis of predictors of PFS in the mCRC cohort.

PFS	Univariate Cox Regression		Multivariable Cox Regression	
	HR (95%CI)	*p*-Value	HR (95%CI)	*p*-Value
Gene stratification				
PRG	1.00		1.00	
AGG	1.91 (1.16–3.15)	0.011	1.84 (1.03–3.28)	0.039
NMG	2.63 (1.26–5.47)	0.010	1.81 (0.65–5.01)	0.253
Age (>65y vs. ≤65y)			1.30 (0.74–2.29)	0.364
Gender (F vs. M)			2.02 (1.11–3.67)	0.021
Grading (G1 vs. G2 vs. G3)			1.05 (0.65–1.70)	0.837
Primay tumor location (RCT vs. RC vs. LC)			0.78 (0.54–1.13)	0.192
Adjuvant therapy (Y vs. N)			1.38 (0.83−2.28)	0.210
Metastatic site (liver and/or other vs. liver)			1.57 (0.92–2.68)	0.100
ECOG PS (≥2 vs. 0–1)			0.22 (0.04–1.17)	0.077
Surgery for primiry tumor (Y vs. N)			0.66 (0.18–2.44)	0.537
Therapy (CHT vs. CHT+antiVEGF vs. CHT+antiEGFR)			0.64 (0.30–1.36)	0.251

PFS = progression-free survival; HR = hazard ratio; CI = confidence interval; F = Female; M = Male; RCT=rectum, RC= right colon; LC=left colon; CHT=chemotherapy.

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
