# Peer review of "A Simplified Genomic Profiling Approach Predicts Outcome in Metastatic Colorectal Cancer"

_cancers, 2019, doi:10.3390/cancers11020147_

Round 1
Reviewer 1 Report
The authors analysed a case series of mCRC patients treated in first line with conventional treatment, and would verify if mutations in a limitated number of genes, evaluated by NGS, were able to predict response to therapy.
Although the intent of the study was interesting, no conclusions could be made from the obtained results due to some weaknesses.
In particular, authors considered all patients together without distinguish the type of treatment (i.e. FOLFOX or FOLFIRI associated with anti-EGFR or anti-VEGF drugs), and they concluded that patients with p53 and/or RAS mutations were those with the best prognosis. However, we known that only RAS-wt patients must be treated with anti-EGFR drugs as RAS mutation is a resistant mechanism to this type of treatment. So, the conclusion that RAS/p53 mutated patients are those more sensitive to therapy, in general, cannot be done. Analysis should be performed separately for the type of treatment and, as a consequence, the case series should be increased to permit to reach a statistical value.
In Table 1 the number of patients for the specific type of treatment should be specified
Moreover, it could be interesting to perform a more accurate analysis with regard to tumor localization (i.e. to evaluate prognosis in right and left-sided tumors separately for RAS wt or RAS mutated tumors)
In the Discussion I think there is a mistake in this paragraph "Additionally, our data indicate that the standard 147 approach has limited efficacy in AGG patients, characterized by the absence of mutations in the genes 148 included in the panel" . The term "AGG" should be changed in "NMG".
The number of references should be improved as is very limited
Author Response
We wish to thank both the Reviewers for their comments on our manuscript, both of which found it interesting.
We have now extensively revised all sections of our paper taking into account all Reviewers’ suggestions. Please, following find our point-by point replies.
Reviewer#1. Point 1
In particular, authors considered all patients together without distinguish the type of treatment (i.e. FOLFOX or FOLFIRI associated with anti-EGFR or anti-VEGF drugs), and they concluded that patients with p53 and/or RAS mutations were those with the best prognosis. However, we known that only RAS-wt patients must be treated with anti-EGFR drugs as RAS mutation is a resistant mechanism to this type of treatment. So, the conclusion that RAS/p53 mutated patients are those more sensitive to therapy, in general, cannot be done. Analysis should be performed separately for the type of treatment and, as a consequence, the case series should be increased to permit to reach a statistical value.
Authors’ Reply to Reviewer#1. Point 1
We acknowledge the Reviewer’s comment and we realized that, for the seek of synthesis, we had poorly described our patients’ cohort, their treatment and their molecular characterization. We have significantly improved all sections of the manuscripts and clarified several aspects. For example, we have now explicitly described the appropriate use of RAS mutations as biomarkers of resistance to anti-EGFR treatment. More in general, we provided information on how each patient was treated in first-line.
Concerning his/her comment “Analysis should be performed separately for the type of treatment“, we wish to point out that extensive literature (summarized in Cremolini et al. Nat Rev Clin Oncol, 2015, and now amply cited in the revised introduction) indicates that the current standard approaches for mCRC (FOLFOX or FOLFIRI on one side, and combination approaches with either anti-EGFR or bevacizumab) are essentially equivalent in PFS. Therefore, as indicated by Cremolini et al, our cohort of patients was managed according to a standard clinical practice, and treated with approaches expected to be equally effective a priori, after taking into account RAS status, individual clinical conditions, and treatment intent.
We believe we can compare PFS after first-line treatment in all of these patients independently of treatment choice, right because they received therapies expected to be equally effective, allowing us to address whether our molecular stratification could predict outcome, in a standard clinical routine. Indeed, in this clinical setting, our stratification in molecular subgroups allowed prediction of better outcome for patients with RAS and/or p53 mutations only (PRG group), and indicated the need for improved understanding and improved treatments for patients with no mutations (NMG) or many mutations (AGG). This is a relevant piece of evidence, with immediate impact on the daily clinical management of mCRC patients, potentially useful to all clinical oncologists.
Interestingly, despite the small size of the cohort, not only our stratification provided significant predictions overall, but did so also in the RAS WT subset of patients, where in principle any of the two biological agents can be chosen. This anticipates that, as soon as we will reach a statistically permissive number of patients, we will also be able to ascertain whether our molecular stratification may predict different responses specifically to anti-EGFR or anti-VEGF treatment in different subgroups.
A substantial increase in the size of the cohort would not be compatible with the timeframe for the revision of the manuscript allowed by the Editor.
Reviewer#1. Point 2
In Table 1 the number of patients for the specific type of treatment should be specified
Authors’ Reply to Reviewer#1. Point 2
Table 1 has been modified according to Reviewer’s request. The number of patients for the specific type of treatment has been provided in materials and methods section.
Reviewer#1. Point 3
Moreover, it could be interesting to perform a more accurate analysis with regard to tumor localization (i.e. to evaluate prognosis in right and left-sided tumors separately for RAS wt or RAS mutated tumors)
Authors’ Reply to Reviewer#1. Point 3
PRG, AGG, NMG stratification revealed no significant correlation with side. Unfortunately the size of our cohort does not allow, at the moment, to further divide the cohort with respect to side and analyze PRG/AGG/NMG stratification with respect to PFS. To answer to Reviewer’s request, however, we found that the side did not provide significant differences at multivariate analysis, while our molecular stratification did (see revised Fig. 3). This is not surprising since a larger clinical sequencing study on CRC samples (Yaeger, et al. 2018) found that molecular alterations, but not tumor location, predicted outcome at a multivariate analysis. As requested by the Reviewer, we have now discussed this issue in the revised discussion section.
Reviewer#1. Point 4
In the Discussion I think there is a mistake in this paragraph "Additionally, our data indicate that the standard 147 approach has limited efficacy in AGG patients, characterized by the absence of mutations in the genes 148 included in the panel" . The term "AGG" should be changed in "NMG".
Authors’ Reply to Reviewer#1. Point 4
This has been corrected according to Reviewer’s suggestion
Reviewer#1. Point 5
The number of references should be improved as is very limited
Authors’ Reply to Reviewer#1. Point 5
This has been corrected according to Reviewer’s suggestion
Reviewer 2 Report
Thank you so much for submitting an interesting paper.
I hope that my comments help you to revise the paper.
1.The title may be inappropriate, because you evaluated survival time after initiation of chemotherapy according to the status of gene mutations, but did not assess the responsiveness to dose-adjusted chemotherapy.
2. Figure 3 and Table 2 can be combined into one Figure, in which adjusted overall survival and progression-free survival would be shown. For adjusment, in addition to age, gender, tumor grade and location, and adjuvant therapy, inclusion of performance status, comorbidity, resection of the primary site, extent of tumor progression, and appropriateness of targeted therapy into multivariate analysis should be considered. As you know, anti-EGFR treatment is recommended for RAS-wild-type tumors, and for RAS-mutant mCRCs anti-VEGF treatment would be considered.
3. To interpret the study results, you are encouraged to explain how the gene mutations you found are involved in deregulation of signaling pathways (WNT, TGF-beta, RTK/RAS, P13K, TP53) in CRC. In addition, please show the possible reason why patients with any gene mutations had worse outcomes.
Author Response
We wish to thank both the Reviewers for their comments on our manuscript, both of which found it interesting.
We have now extensively revised all sections of our paper taking into account all Reviewers’ suggestions. Please, following find our point-by point replies.
Reviewer#2. Point 1
The title may be inappropriate, because you evaluated survival time after initiation of chemotherapy according to the status of gene mutations, but did not assess the responsiveness to dose-adjusted chemotherapy.
Authors’ Reply to Reviewer#2. Point 1
The title has been modified according to Reviewer’s suggestion
Reviewer#2. Point 2
Figure 3 and Table 2 can be combined into one Figure, in which adjusted overall survival and progression-free survival would be shown. For adjustment, in addition to age, gender, tumor grade and location, and adjuvant therapy, inclusion of performance status, comorbidity, resection of the primary site, extent of tumor progression, and appropriateness of targeted therapy into multivariate analysis should be considered. As you know, anti-EGFR treatment is recommended for RAS-wild-type tumors, and for RAS-mutant mCRCs anti-VEGF treatment would be considered.
Authors’ Reply to Reviewer#2. Point 2
Figure 3 has been revised according to Reviewer’s request. Adjustments have also been made according to Reviewer’s request and multivariate analysis now includes the variables indicated by the Reviewer.
With reference to Overall Survival, which very much depends also on locoregional treatments and subsequent lines of therapy, among a large number of other variables, we thought it could not be an endpoint to be satisfactory met in our relatively small cohort of patients. Indeed, the endpoint of the study was to establish whether the different patterns of genetic alterations impact on the outcome, when first-line treatment choice was dictated only by RAS status (as the only “molecular variable”), in addition to clinical features of the patients and treatment intent. Of note, our work clearly supports the message that RAS status alone (and by extension we believe any “single gene assessment”) may not be enough to indicate the most appropriate therapeutic strategy.
Reviewer#2. Point 3
To interpret the study results, you are encouraged to explain how the gene mutations you found are involved in deregulation of signaling pathways (WNT, TGF-beta, RTK/RAS, P13K, TP53) in CRC. In addition, please show the possible reason why patients with any gene mutations had worse outcomes.
Authors’ Reply to Reviewer#1. Point 3
The work by others has extensively addressed the oncogenic activation of several pathways in CRC, by global and integrated NGS-based genomic and transcriptomic profiling, which have led to the description of the pathways most frequently activated in CRC and to the definition of a molecular classification system (Guinney, et al 2015; Cancer Genome Atlas, 2012). Prompted by the Reviewer’s comment we have now briefly discussed this notion in the discussion section.
Importantly, these approaches cannot be easily implemented in a standard clinical routine, because of costs and even more because treatment choice needs to be done in short times, not readily compatible with detailed and integrated molecular studies such as those described above. In our studies, instead we looked at the mutational status of 22 genes, with a method that provides costs and turnaround time perfectly compatible with the clinical routine. The data we managed cannot allow us to draw any conclusions in terms of “pathway analysis”. Rather, they can inform us about the significance of molecular patterns. Indeed, we used an unsupervised (and thus unbiased) analysis to cluster our patient cohort into molecular subgroups and shown this impacts on PFS in response to first-line therapy. Therefore we took into account the Reviewer’s suggestion and extensively discussed what we think are the reasons for such a different behaviour in the three molecular subgroups we identified, in this extensively revised version of the manuscript.
Round 2
Reviewer 1 Report
The authors have extensively changed the manuscript that could now be accepted for publication.
Reviewer 2 Report
I appreciate your efforts to revise the manuscript, and congratulate your success.